# Comparison of Short-Term Surgical Outcomes According to Immediately Postoperative Serum Glucose Level in Non-Diabetic Pancreatic Resection Patients

**DOI:** 10.3390/biomedicines10102427

**Published:** 2022-09-28

**Authors:** Okjoo Lee, Chang-Sup Lim, So Jeong Yoon, Ji Hye Jung, Sang Hyun Shin, Jin Seok Heo, Yong Chan Shin, Woohyun Jung, In Woong Han

**Affiliations:** 1Department of Surgery, Soonchunhyang University Bucheon Hospital, Soonchunhyang University College of Medicine, 170, Jomaru-ro, Bucheon 14584, Korea; 2Department of Surgery, Seoul Metropolitan Government-Seoul National University Boramae Medical Center, Seoul National University College of Medicine, 20, Boramae-ro 5-gil, Dongjak-gu, Seoul 07061, Korea; 3Division of Hepatobiliary-Pancreatic Surgery, Department of Surgery, Samsung Medical Center, Sungkyunkwan University College of Medicine, 81 Irwon-ro, Gangnam-gu, Seoul 06351, Korea; 4Department of Surgery, Ilsan Paik Hospital, Inje University College of Medicine, Juhwa-ro 170, Ilsanseo-gu, Goyang 10380, Korea; 5Department of Surgery, Ajou University Hospital, Ajou University College of Medicine, 164, World Cup-ro, Yeongtong-gu, Suwon 16499, Korea

**Keywords:** pancreatectomy, glycemic control, postoperative complications

## Abstract

The adequate regulation of postoperative serum glucose level (SGL) is widely accepted; however, the effects for non-diabetic patients who underwent major pancreatic surgery have not yet been established. We discerned the relevance of the immediately postoperative SGL to short-term postoperative outcomes from major pancreatic surgery in non-diabetic patients. Between January 2007 and December 2016, 2259 non-diabetic patients underwent major pancreatic surgery at four tertiary medical centers in Republic of Korea. Based on a SGL of 200 mg/dL, patients were classified into two groups by averaging the results of four SGL tests taken on the first day after surgery, and their short-term postoperative outcomes were analyzed. A 1:1 propensity score matching method was conducted to establish the high SGL group (*n* = 568) and the normal SGL group (*n* = 568). The high SGL group experienced a significantly higher rate of level C complications in the Clavien-Dindo classification (CDc) than the normal SGL group (24.1% vs. 16.5%, *p* = 0.002). Additionally, an SGL of more than 200 mg/dL was associated with a significantly high risk of complications above level C CDc after adjusting for other risk factors (hazard ratio = 1.324, 95% confidence interval = 1.048–1.672, *p* = 0.019). The regulation of SGL of less than 200 mg/dL in non-diabetic patients early after major pancreatic surgery could be helpful for reducing postoperative complications.

## 1. Introduction

Adequate regulation of postoperative blood glucose has been linked with better prognosis in many studies, and it is becoming widely accepted as a standard patient management strategy for general surgery, including pancreatic surgery. In the past, the effects of postoperative blood glucose management on patient prognosis have been studied in intensive care unit patients, and patients with appropriate blood glucose levels have had better outcomes [1,2,3]. In subsequent studies of patients who underwent general surgery, an increase in blood sugar correlated with an increase in postoperative complications and infections, the length of hospital stay, and hospital costs [4,5,6]. In addition, the guidelines recommend that blood sugar be controlled before and after surgery to a target blood sugar level of less than 200 mg/dL for patients with and without diabetes mellitus (DM; Category IA—strong recommendation; high-to-moderate quality evidence) [7]. However, our literature search did not find any randomized controlled trials that evaluated lower (<200 mg/dL) or narrower blood glucose target levels than those recommended in that guideline, nor did we find studies showing the optimal timing, duration, or delivery method for perioperative glycemic control for the prevention of surgical site infections (SSIs).

We did find several studies on the relationship between pancreatic surgery and the control of the serum glucose level (SGL). One study showed that a postoperative SGL of above 200 mg/dL increased the incidence of SSIs after surgery for hepato-biliary-pancreatic cancer [8]. In another study, an early postoperative glucose level of higher than 140 mg/dL was significantly associated with postoperative complications in patients with and without DM [9]. The intensive SGL control group had significantly fewer bile leaks, pancreatic fistulae, and hospitalizations than the intermediate glucose control group, in an artificial endocrine pancreas study for patients who received a pancreatic resection; complications with a Clavien-Dindo classification (CDc) of higher than grade III occurred significantly less in the group whose target was 120–180 mg/dL [10,11]. Other studies have shown that an artificial pancreas promotes tight and safe glycemic control while reducing anti-inflammatory mediators, including adiponectin, after pancreaticoduodenectomy (PD) [12].

In general, patients with underlying DM already have a clinical strategy for glucose control (Alberti’s regimen and rapid-acting insulin), and it is common to treat each patient with their own previous treatment method [13,14]. In most cases, clinicians pay careful attention to glucose control in patients with diabetes. However, in patients without DM, clinicians respond passively with SGL change. Often, only blood sugar test monitoring is performed, and no action is usually taken, even when the blood sugar level is high. Of course, patients without diabetes generally have relatively good blood sugar control. However, among patients with subclinical diabetes, such as those with impaired glucose tolerance, changes in blood sugar can be masked in unstable postoperative conditions. Additionally, although most researchers agree that diabetes is a risk factor for surgical complications, they have not separately analyzed patients without diabetes.

The pancreas is the main regulatory organ for glucose metabolism. Decreased pancreas function following major pancreatic surgery can complicate glucose metabolism, even in non-diabetic patients. Therefore, a decrease in pancreatic function, combined with the clinical problems mentioned above, could cause an increase in morbidity, even in non-diabetic patients.

In this study, we investigated the relevance of the immediately postoperative blood glucose level to the short-term postoperative outcomes of non-diabetic patients who underwent major pancreatic surgery.

## 2. Materials and Methods

### 2.1. Patients and Study Outline

From January 2007 to December 2016, a multi-institutional retrospective study was conducted on 3108 patients who underwent major pancreatic resection for pancreatic tumors at Samsung Medical Center (Seoul, Korea), Seoul National University Boramae Medical Center (Seoul, Korea), Ajou University School of Medicine (Suwon, Korea), and Inje University Ilsan Paik Hospital (Ilsan, Korea) (Figure 1). Patients younger than 19 years of age and those with other major surgeries, suspected metastatic or multiple primary cancers, or a diagnosis of diabetes, were excluded (*n* = 831). Patients with missing data or lost to follow-up were also excluded (*n* = 18). The remaining 2259 eligible patients were included in this study analysis.

We divided the patients into 2 groups, those whose average SGL was below 200 mg/dL on postoperative day 1 (nSGL group, *n* = 1691), and those whose average SGL was more than 200 mg/dL on postoperative day 1 (hSGL group, *n* = 568). After the group classification, we conducted 1:1 propensity score matching to minimize the selection bias inherent in any retrospective analysis. We compared clinical and laboratory data, and proceeded to analyze risk factors for postoperative complications. Every step of this study was carried out after receiving the approval of the Institutional Review Board (SMC 2018-05-079, 16.05.2018 approved).

### 2.2. Surgical Procedures for Major Pancreatic Surgery

Major pancreatic surgery was defined as PD and distal pancreatectomy (DP). The standard surgery for a pancreatic head lesion is pylorus-preserving PD with lymph node dissection, as earlier described [15]. When a tumor infiltrated the pylorus or duodenum, a classic Whipple’s operation was performed. If minimal tumor invasion of the portal or superior mesenteric vein was found, a segmental or tangential resection was conducted [16].

Retrocolic hepaticojejunostomy, pancreaticojejunostomy, and antecolic duodenojejunostomy were used as standard reconstruction methods. Two Jackson-Pratt drains were left in the abdominal cavity, one in the foramen of Winslow near the hepaticojejunostomy, and the other in the posterior area of the pancreaticojejunostomy. Postoperatively, these patients stayed at the surgical intensive care unit until the morning of the first postoperative day, and then returned to the surgical ward.

For body and tail lesions that were benign disease, DP (spleen preserving or not) was performed, and for those that were malignant disease, a radical antegrade modular pancreatosplenectomy (anterior or posterior) was carried out [17,18]. The portal dissection includes clearance of the portal vein and the hepatic artery lymph nodes, and a celiac lymphadenectomy was also performed. One Jackson-Pratt drain was left in the posterior area of the stomach near the stump of the pancreatic resection, and these patients returned directly to the surgical ward postoperatively.

### 2.3. Definition and Management of Hyperglycemia

The current consensus diagnostic glycemic criteria for pre-symptomatic diabetes are: (1) glycosylated hemoglobin A1c (HbA1c) ≥ 6.5%; (2) fasting SGL ≥ 126 mg/dL or SGL measured 2 h after an oral glucose tolerance test (2 h SGL) ≥ 200 mg/dL [19,20]. For patients with typical symptoms, a random SGL ≥ 200 mg/dL is diagnostic. Generally speaking, the 2 h SGL yields the highest prevalence, and HbA1c, the lowest [21].

Based on those diagnostic criteria, we searched related studies to define postoperative hyperglycemia prior to our full-scale analysis [7,8]. We checked the area under the receiver operating characteristic (ROC) curve (AUC) values by drawing ROC curves for SGL and the main outcomes of the entire cohort, and we set the criterion for postoperative hyperglycemia at SGL ≥ 200 mg/dL.

The management of postoperative hyperglycemia depends on the frequency of the SGL measurement; we tested the SGL every 6 h until postoperative day 1, regardless of whether the patient had DM. We calculated the immediately postoperative and mean postoperative day 1 SGL for each patient. If a patient had an SGL ≥ 200 mg/dL, we immediately administered short-acting insulin subcutaneously. The standard used for insulin administration was as follows: for patients with blood glucose levels of 200–250 mg/dL or 250–300 mg/dL, we subcutaneously injected 4–8 IU or 8–12 IU of short-acting insulin, respectively. If patients had difficult or intermittent control of SGL, we postoperatively administered short-acting insulin using a continuous intravenous infusion (1–2 IU/h).

Sips of water were attempted in the morning of postoperative day 1 in every patient, regardless of what kind of surgery was performed. Enteral feeding proceeded as soon as patients were hemodynamically stable.

### 2.4. Definitions of Variables and Outcome Measures

Postoperative morbidity and mortality were noted during hospital admission, and 30 days after discharge. The definition and grade of postoperative complications used the CDc [22]. A CDc grade ≥ III was considered as a major complication.

All patients were checked daily for signs of infection; the diagnosis of infection was confirmed by a positive bacteriological culture. According to the Centers for Disease Control and Prevention (CDC), all SSIs have at least one of the following: (1) purulent discharge from the superficial incision with or without laboratory confirmation; (2) organisms isolated from an aseptically obtained culture of fluid or tissue from the superficial incision [23].

All patients received antibiotic prophylaxis for three days after surgery. We took preventive measures against SSIs, in accordance with CDC guidelines [23]. Our policy was that the physician in charge at each hospital directly examined patients’ wounds during the hospitalization period, and observed each patient every 3 months for 1 year after surgery, beginning 2 weeks after discharge.

### 2.5. Statistical Analysis

All variables are expressed as the mean ± standard deviation, or the number and percentage. Between-group differences in the mean values were compared with independent *t*-tests, and between-group differences in the numbers and percentages were compared with Chi-squared tests or Fisher’s exact test. Kaplan–Meier survival curves were used to calculate complication-related survival and to evaluate the regulation of SGL. A logistic regression analysis was performed with adjustments for risk factors that were significant in the univariate analysis, as well as other factors that were previously known to be associated with major complications. All *p*-values of less than 0.05 were considered as statistically significant. All statistical analyses were performed using SPSS (version 25; IBM Corp., Armonk, NY, USA) and R (version 4.2.0; The R Foundation for Statistical Computing, Vienna, Austria) software.

## 3. Results

### 3.1. Baseline Characteristics and Primary Outcomes

We reviewed data from 2259 patients who underwent a major pancreatic resection between January 2007 and December 2016. The nSGL group (*n* = 1691) and hSGL group (*n* = 568) were compared, to find any differences in their demographic and perioperative characteristics.

The hSGL group was significantly older than the nSGL group (63.36 ± 11.44 vs. 59.58 ± 12.62, *p <* 0.001), had a higher prevalence of hypertension (36.4% vs. 29.2%, *p =* 0.001), and higher preoperative serum HbA1c level (Table 1). The proportions of male patients (48.6% vs. 57.3%, *p <* 0.001) and those who underwent PD (70.2% vs. 74.6%, *p =* 0.046) were higher in the nSGL group than the hSGL group. We performed 1:1 propensity score matching to balance the differences in variables (Sex, age, hypertension, and operation type) between the two groups. After propensity score matching, the matched baseline demographic and clinical variables did not differ significantly, except for age and pre-operative ASA score.

Hospital stay (16.10 ± 11.99 vs. 15.30 ± 22.07, *p =* 0.825) was longer in the hSGL group than the nSGL group (Table 2). Operation time and estimated blood loss did not differ significantly, and the pathologic outcomes did not differ between the groups. The overall postoperative complication rate was similar between the two groups, but the hSGL group showed a significantly higher postoperative 1-day average SGL (231.2 ± 32.3 vs. 165.2 ± 20.1, *p <* 0.001), rate of complications above CDc grade III (24.1% vs. 19.8%, *p =* 0.031), and proportion of patients requiring re-operation (4.2% vs. 2.5%, *p =* 0.047).

### 3.2. Risk Factor Analysis for Complications

The cumulative major complication rate was estimated using a Kaplan–Meier analysis (Figure 2). The Kaplan–Meier curve showed that major complication-related survival was significantly higher in the nSGL group than in the hSGL group (*p =* 0.002). The cumulative re-operation-related analysis also differed significantly between the groups (*p =* 0.010; Figure 2).

The risk factor analysis for major complications used logistic regression models (Table 3). We selected variables that were universal risk factors of major complication and main outcomes of this study. Being male (hazard ratio [HR] = 1.439, 95% confidence interval [CI] = 1.162–1.782, *p <* 0.001), age ≥ 60 years (HR = 1.371, 95% CI = 1.104–1.703, *p =* 0.004), BMI >25 (HR = 1.335, 95% CI = 1.067–1.670, *p =* 0.011), and having chronic kidney disease (HR = 3.719, 95% CI = 1.435–9.637, *p =* 0.007), PD (HR = 1.876, 95% CI = 1.438–2.447, *p <* 0.001), postoperative 1-day average SGL ≥ 200 mg/dL (HR = 1.324, 95% CI = 1.048–1.672, *p =* 0.019), or immediately postoperative SGL ≥ 200 mg/dL (HR = 1.267, 95% CI = 1.019–1.574, *p =* 0.033) all had a significant relationship with major complications after major pancreatic resection.

The significant risk factors for re-operation were being male (HR = 2.053, 95% CI = 1.191–3.539, *p =* 0.010) and having hypertension (HR = 1.977, 95% CI = 1.181–3.309, *p =* 0.010) or a postoperative 1-day average SGL ≥ 200 mg/dL (HR = 1.638, 95% CI = 1.007–2.757, *p =* 0.048). An immediately postoperative SGL ≥ 200 mg/dL was not significantly related to prognosis (HR = 1.214, 95% CI = 0.726–2.030, *p =* 0.461) (Table 4).

The risk factors for re-admission were having chronic kidney disease (HR = 3.274, 95% CI = 1.150–9.325, *p =* 0.026), neoadjuvant chemotherapy (HR = 2.951, 95% CI = 1.164–7.476, *p =* 0.023), PD (HR = 2.043, 95% CI = 1.403–2.974, *p <* 0.001), or an immediately postoperative SGL ≥ 200 mg/dL (HR = 1.433, 95% CI = 1.076–1.910, *p =* 0.014). A postoperative 1-day average SGL ≥ 200 mg/dL was not significantly related to prognosis (HR = 0.765, 95% CI = 0.543–1.077, *p =* 0.125) (Table 5).

## 4. Discussion

Many studies have shown that postoperative SGL control could be helpful in reducing postoperative complications, including SSIs, and hyperglycemia or diabetes is widely established as a risk factor in major abdominal surgery [7,8,9]. However, relatively few studies have investigated the effect of hyperglycemia on major complications in patients undergoing major pancreatic surgery, particularly non-diabetics. Therefore, here we have targeted only non-diabetic patients who received major pancreatic surgery. Obviously, diabetes is the most important underlying disease for glycemic control, and the pancreas is the major organ for glycemic metabolism. Therefore, we expected that the relationship between major pancreatic surgery and SGL in non-diabetic patients would show more homogenous results for the general population than in previous studies of other types of surgery.

The diagnostic criteria for diabetes were used to establish the criteria for postoperative hyperglycemia in this study [19,20]. The main outcome of this study was complications according to the SGL on the first day after surgery; the dietary progress after surgery was not considered here. Because the AUC value in the ROC curve for setting the blood glucose standard was low (AUC 0.516), it was difficult to fit as a diagnostic standard. Therefore, we set the standard by grafting the clinical diabetes diagnostic criteria for randomized blood glucose, SGL ≥ 200 mg/dL. Although we did not report the results, we analyzed our data using criteria such as SGL > 140 mg/dL (the standard of the previous study) and SGL > 170 mg/dL (the average of the entire cohort in this study) [9]. Those analyses produced no significant differences in the incidence of complications in diabetic and non-diabetic patients.

Several studies have applied stricter criteria than we did, and their results associating hyperglycemia with poor prognosis were similar to ours [2,10]. However, diabetic patients were enrolled in those study groups. Of course, in our study, it is difficult to present the exact criteria for postoperative SGL or for the pathogenesis of complications. Other results could be caused by the decreased sensitivity of body tissues to glucose attacks, due to chronic hyperglycemia in diabetic patients. However, we analyzed a relatively large number of patients and reduced heterogeneity by excluding patients diagnosed with diabetes. In that way, we obtained the result that a postoperative 1-day average SGL ≥ 200 mg/dL after major pancreatic surgery in non-diabetic patients correlates with the occurrence of major complications.

In this study, major pancreatic resections were limited to PD and DP. Although we analyzed them together, it is clear that PD is a more burdensome operation for the patient than DP. Indeed, we found that PD was a significant risk factor for major complications and re-admission. In general, the complication rates following PD are 20% to 30% [24,25,26]. In our data, CDc grade ≥ III major complications occurred in 17.4% of PD patients, and the re-admission and re-operation rates were 11.3% and 3.1%, respectively [27]. Nonetheless, our hSGL group had a higher rate of major complications, despite having a smaller proportion of PD patients (24.1% vs. 19.8%). This suggests that systemic SGL management can be as important as the scale of pancreatic surgery. We did not find any significant difference between the nSGL and hSGL groups in the occurrence of postoperative pancreatic fistula (POPF), a postoperative outcome that can only occur in pancreatic resection patients (*p =* 0.238). This might simply mean that there is no association between hyperglycemia and POPF in non-diabetic patients. Because few studies have considered the risk factors for POPF in non-DM patients, additional studies are needed.

Because the results of this study are based on data collected immediately after surgery, we presume that they significantly reflect preoperative conditions. Therefore, the results of this study indicate that it is necessary to evaluate preoperative SGL control as part of clinical patient management. Moreover, measures should be prepared to increase preoperative glucose tolerance and preoperative SGL control. Recently, several studies have reported on prehabilitation for sarcopenia and sarcopenic obesity, placing those conditions in the spotlight as preoperative risk factors [28,29]. SGL control and glucose tolerance are closely correlated with sarcopenia because insulin receptors in the muscle play a major role in glucose regulation, and muscles are a major site of glucose disposal. Therefore, poor glycemic control in patients is associated with low muscle mass [30]. Because most pancreas resection patients are elderly, hyperglycemia, sarcopenia, and cognitive dysfunction are mutually associated in complex ways. Therefore, an understanding of the mechanisms that underlie those associations is needed to devise effective strategies for preventing and treating this patient population [31].

In addition, clinical evidence supports the implementation of a preoperative management protocol of physical exercise to reduce major postoperative complications in high-risk patients undergoing a major abdominal operation [32]. Moreover, body composition parameters should routinely be assessed, including prehabilitation programs for high-risk sarcopenic patients. Sarcopenic patients with low preoperative physical status who are at high risk for major complications might benefit from a prehabilitation strategy that focuses on chronic morbidity management, nutrition, and physical activity status [33,34]. Improving glucose tolerance through preoperative prehabilitation thus has the potential to reduce the incidence of complications.

There were some limitations to this study. First, it was a retrospective and multicenter analysis, and each center has different treatment strategies for SGL control. Second, in defining postoperative hyperglycemia in patients not diagnosed with diabetes, individual functional aspects were not sufficiently estimated, due to the retrospective study design. Using the diagnostic criteria for diabetes, we set 200 mg/dL as the standard for postoperative hyperglycemia, but no clear clinical consensus on that criterion has been reached. The AUC value of the ROC curve was low, so we set the diagnostic standard as a clinical criterion. However, to the best of our knowledge, it is the first study to report the adequate regulation of postoperative SGL with major pancreatic resection in non-diabetic patients. Therefore, through this study, it may be possible to contribute to non-diabetic patient management, even with the risk of complication with high SGL.

In conclusion, keeping the SGL for non-diabetic patients to less than 200 mg/dL, 1 day after major pancreatic surgery could be helpful in reducing major postoperative complications. By the same token, a postoperative SGL of more than 200 mg/dL is a poor predictive factor for re-operation. Our findings indicate that perioperative SGL management can make a major pancreatic resection a safe and feasible option for treating pancreatic lesions in non-diabetic patients. However, many issues remain to be addressed, such as optimal target blood sugar levels and monitoring intervals. A large-scale, prospective, randomized study is needed to further investigate the long-term effects of postoperative hyperglycemia in non-diabetic patients. We expect that a careful and definitive evaluation of perioperative hyperglycemia will increase patient survival and improve the prognosis of non-diabetic patients.

## Figures and Tables

**Figure 1 biomedicines-10-02427-f001:**
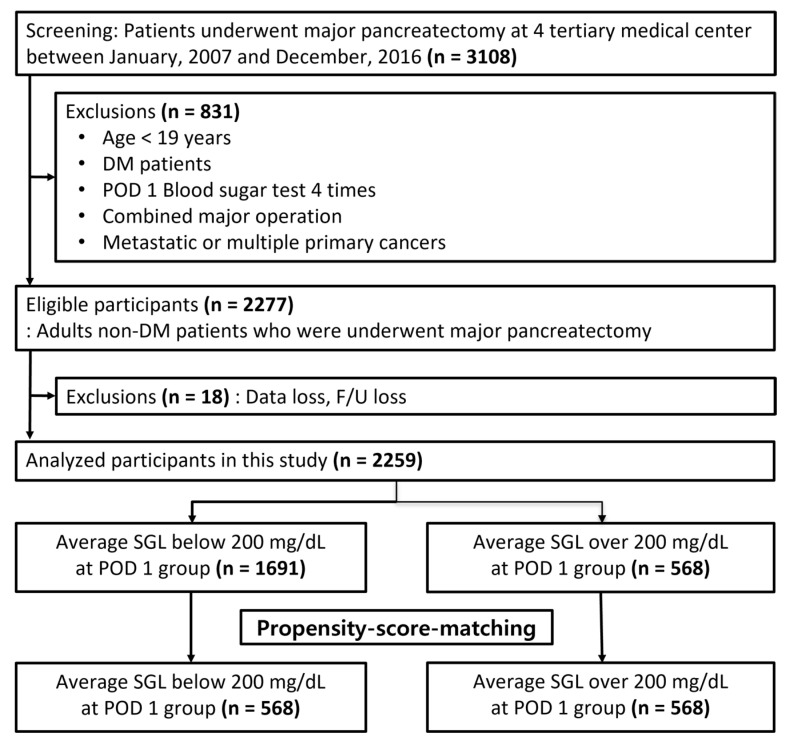
Flow diagram of patient selection.

**Figure 2 biomedicines-10-02427-f002:**
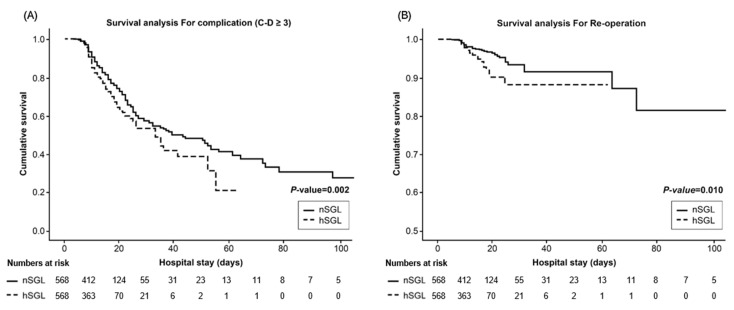
Cumulative major complications (**A**) and re-operations (**B**) estimated using the Kaplan–Meier method.

**Table 1 biomedicines-10-02427-t001:** Preoperative characteristics.

Characteristics	All Patients			Propensity-Score-Matched Patients		
	SGL < 200 (*n* = 1691)	SGL ≥ 200 (*n* = 568)	*p*-Value	SGL < 200 (*n* = 568)	SGL ≥ 200 (*n* = 568)	*p*-Value
Sex (male)	969 (57.3)	276 (48.6)	<0.001	276 (48.6)	276 (48.6)	1.000
Age (years)	59.58 ± 12.62	63.36 ± 11.44	<0.001	61.62 ± 12.19	63.36 ± 11.44	0.013
Body mass index	23.26 ± 3.17	23.36 ± 3.31	0.553	23.21 ± 3.13	23.36 ± 3.31	0.430
Hypertension	493 (29.2)	207 (36.4)	0.001	207 (36.4)	207 (36.4)	1.000
Chronic kidney disease	13 (0.8)	5 (0.9)	0.787	2 (0.4)	5 (0.9)	0.452
Pre op. ASA score			0.128			0.034
I	522 (31.1)	160 (28.2)		180 (31.7)	160 (28.2)	
II	1068 (63.5)	366 (64.4)		367 (64.6)	366 (64.4)	
III	89 (5.3)	41 (7.2)		20 (3.5)	41 (7.2)	
IV	1 (0.1)	1 (0.2)		0 (0.0)	1 (0.2)	
V	1 (0.1)	0 (0.0)		1 (0.2)	0 (0.0)	
Pre op. lab findings						
Albumin	4.04 ± 0.58	4.04 ± 0.51	0.831	4.02 ± 0.47	4.04 ± 0.51	0.658
Serum glucose level	116.06 ± 40.9	115.81 ± 37.1	0.894	112.24 ± 38.3	115.81 ± 37.1	0.793
HbA1c	5.61 ± 0.89	5.86 ± 0.99	0.008	5.56 ± 0.74	6.04 ± 1.09	<0.001
CEA	2.48 ± 4.06	2.76 ± 5.68	0.379	2.69 ± 5.68	2.76 ± 5.68	0.467
CA 19-9	327.81 ± 1156.8	318.21 ± 1085.8	0.863	328.28 ± 1014.8	318.21 ± 1085.8	0.730
Neoadjuvant CTx	17 (1.0)	10 (1.8)	0.226	4 (0.7)	10 (1.8)	0.134
Operation type			0.046			1.000
PD	1262 (74.6)	399 (70.2)		399 (70.2)	399 (70.2)	
DP	429 (25.4)	169 (29.8)		169 (29.8)	169 (29.8)	

Pre op., preoperative; ASA, American Society of Anesthesiologists; HbA1c, hemoglobin A1c; CEA, carcinoembryonic antigen; CA 19-9, carbohydrate antigen 19-9; CTx, chemotherapy; PD, pancreaticoduodenectomy; DP, distal pancreatectomy; SGL, serum glucose level.

**Table 2 biomedicines-10-02427-t002:** Postoperative outcomes.

Characteristics	All Patients			Propensity-Score-Matched Patients		
	SGL < 200 (*n* = 1691)	SGL ≥ 200 (*n* = 568)	*p*-Value	SGL < 200 (*n* = 568)	SGL ≥ 200 (*n* = 568)	*p*-Value
EBL	478.57 ± 510.1	453.95 ± 354.3	0.205	414.00 ± 283.29	453.95 ± 354.3	0.135
Op. duration	294.82 ± 92.8	289.53 ± 95.1	0.249	273.15 ± 84.9	289.53 ± 95.1	0.526
Hospital stay	15.30 ± 22.07	16.10 ± 11.99	0.825	12.93 ± 7.01	16.10 ± 11.99	<0.001
Tumor type			0.658			0.394
PDAC	1100 (65.1)	357 (62.9)		336 (59.2)	357 (62.9)	
PNET	81 (4.8)	32 (5.6)		40 (7.0)	32 (5.6)	
IPMN	166 (9.8)	63 (11.1)		69 (12.1)	63 (11.1)	
Other	344 (20.3)	116 (20.4)		123 (21.7)	116 (20.4)	
Pathologic findings						
Tumor size	3.10 ± 2.11	3.20 ± 2.00	0.319	3.11 ± 1.83	3.20 ± 2.00	0.435
LN metastasis	1.31 ± 2.73	1.33 ± 2.84	0.897	1.34 ± 3.01	1.33 ± 2.84	0.839
Lymphovascular invasion	530 (31.3)	153 (26.9)	0.054	160 (28.2)	153 (26.9)	0.895
Perineural invasion	745 (44.1)	252 (44.4)	0.937	221 (38.9)	252 (44.4)	0.430
Post op. initial SGL	184.9 ± 49.1	182.0 ± 46.8	0.210	177.69 ± 45.7	182.0 ± 46.8	0.566
Post op. 1-day average SGL	165.2 ± 20.1	231.2 ± 32.3	<0.001	172.7 ± 16.9	231.2 ± 32.3	<0.001
Complications (CDc)			0.277			0.012
0	831 (49.1)	272 (47.9)		286 (50.4)	272 (47.9)	
I	211 (12.5)	68 (12.0)		83 (14.6)	68 (12.0)	
II	315 (18.6)	91 (16.0)		105 (18.5)	91 (16.0)	
III	274 (16.2)	111 (19.5)		84 (14.8)	111 (19.5)	
IV	37 (2.2)	14 (2.5)		7 (1.2)	14 (2.5)	
V	23 (1.4)	12 (2.1)		3 (0.5)	12 (2.1)	
Severe complications (CDc ≥ Grade III)	334 (19.8)	137 (24.1)	0.031	94 (16.5)	137 (24.1)	0.002
POPF (≥Grade B)	179 (10.6)	71 (12.5)	0.238	63 (11.1)	71 (12.5)	0.520
Re-operation	42 (2.5)	24 (4.2)	0.047	14 (2.5)	24 (4.2)	0.137
Re-admission	177 (10.5)	46 (8.1)	0.120	43 (7.6)	46 (8.1)	0.825
Death	34 (2.0)	7 (1.2)	0.308	4 (0.7)	7 (1.2)	0.131

EBL, estimated blood loss; Op., operative; PDAC, pancreatic ductal adenocarcinoma; PNET, pancreatic neuroendocrine tumor; IPMN, intraductal papillary mucinous neoplasm; LN, lymph node; SGL, serum glucose level; Post op., postoperative; CDc, Clavien-Dindo classification; POPF, postoperative pancreatic fistula.

**Table 3 biomedicines-10-02427-t003:** Logistic regression for risk factor analysis of major complications.

Characteristics	Univariable		Multivariable	
	HR (95% CI)	*p*-Value	HR (95% CI)	*p*-Value
Sex (male)	1.512 (1.227–1.864)	<0.001	1.439 (1.162–1.782)	<0.001
Old age (≥60 years)	1.539 (1.247–1.901)	<0.001	1.371 (1.104–1.703)	0.004
High BMI (≥25)	1.284 (1.031–1.599)	0.026	1.335 (1.067–1.670)	0.011
Hypertension	1.203 (0.970–1.493)	0.092		
Chronic kidney disease	4.826 (1.894–12.300)	<0.001	3.719 (1.435–9.637)	0.007
Neoadjuvant CTx	1.334 (0.561–3.173)	0.515		
PD	1.993 (1.535–2.589)	<0.001	1.876 (1.438–2.447)	<0.001
PDAC	1.115 (0.900–1.382)	0.319		
POD 1-day avg. SGL (≥200 mg/dL)	1.291 (1.030–1.620)	0.027	1.324 (1.048–1.672)	0.019
High post op. SGL (≥200 mg/dL)	1.270 (1.026–1.572)	0.028	1.267 (1.019–1.574)	0.033

BMI, body mass index; CTx, chemotherapy; PD, pancreaticoduodenectomy; PDAC, pancreatic ductal adenocarcinoma; Op., operative; POD, postoperative day; avg., average; SGL, serum glucose level; Post op., postoperative; HR, hazard ratio; CI, confidence interval.

**Table 4 biomedicines-10-02427-t004:** Logistic regression for risk factor analysis of re-operation.

Characteristics	Univariable		Multivariable	
	HR (95% CI)	*p*-Value	HR (95% CI)	*p*-Value
Sex (male)	2.055 (1.198–3.523)	0.009	2.053 (1.191–3.539)	0.010
Old age (≥60 years)	2.433 (1.377–4.299)	0.002	1.774 (0.973–3.234)	0.061
High BMI (≥25)	0.882 (0.504–1.544)	0.661		
Hypertension	2.436 (1.491–3.981)	<0.001	1.977 (1.181–3.309)	0.010
Chronic kidney disease	4.252 (0.957–18.883)	0.057	2.530 (0.555–11.533)	0.230
Neoadjuvant CTx	1.282 (0.171–9.594)	0.809		
PD	1.641 (0.872–3.089)	0.125		
PDAC	0.843 (0.510–1.391)	0.503		
POD 1-day avg. SGL (≥200 mg/dL)	1.732 (1.039–2.887)	0.035	1.638 (1.007–2.757)	0.048
High post op. SGL (≥200 mg/dL)	1.239 (0.745–2.063)	0.409	1.214 (0.726–2.030)	0.461

BMI, body mass index; CTx, chemotherapy; PD, pancreaticoduodenectomy; PDAC, pancreatic ductal adenocarcinoma; Op., operative; POD, postoperative day; avg., average; SGL, serum glucose level; Post op., postoperative; HR, hazard ratio; CI, confidence interval.

**Table 5 biomedicines-10-02427-t005:** Logistic regression for risk factor analysis of re-admission.

Characteristics	Univariable		Multivariable	
	HR (95% CI)	*p*-Value	HR (95% CI)	*p*-Value
Sex (male)	1.280 (0.965–1.697)	0.087		
Old age (≥60 years)	1.071 (0.809–1.417)	0.634		
High BMI (≥25)	1.161 (0.860–1.567)	0.330		
Hypertension	1.021 (0.758–1.376)	0.891		
Chronic kidney disease	3.569 (1.260–10.106)	0.017	3.274 (1.150–9.325)	0.026
Neoadjuvant CTx	2.653 (1.059–6.644)	0.037	2.951 (1.164–7.476)	0.023
PD	2.053 (1.413–2.983)	<0.001	2.043 (1.403–2.974)	<0.001
PDAC	1.026 (0.768–1.371)	0.863		
POD 1-day avg. SGL (≥200 mg/dL)	0.754 (0.537–1.058)	0.103	0.765 (0.543–1.077)	0.125
High post op. SGL (≥200 mg/dL)	1.431 (1.076–1.903)	0.014	1.433 (1.076–1.910)	0.014

BMI, body mass index; CTx, chemotherapy; PD, pancreaticoduodenectomy; PDAC, pancreatic ductal adenocarcinoma; Op., operative; POD, postoperative day; avg., average; SGL, serum glucose level; Post op., postoperative; HR, hazard ratio; CI, confidence interval.

## Data Availability

All data generated in this study are available from the corresponding author upon request.

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
