# Peer review of "Comparison of Short-Term Surgical Outcomes According to Immediately Postoperative Serum Glucose Level in Non-Diabetic Pancreatic Resection Patients"

_biomedicines, 2022, doi:10.3390/biomedicines10102427_

Round 1

Reviewer 1 Report

Lee et al., shows that regulating serum glucose levels below 200 mg/dL potentially reduces the risk of post operative complications after major pancreatic surgery in non-diabetic patients. This work provides a valuable new data that may inform clinical management of patients undergoing pancreatic surgery. Overall, the paper is well presented except for minor required changes. Please see below.

Minor comments

Please include only relevant data in the tables. Any numbers that are not directly relevant to the study findings can be removed or put in the supplemental tables.

Figure 2- figures can be labeled as figure 2A and figure 2B

Please make sure to avoid odd phrases such as ‘smooth treatment’ (line 73), ‘clinicians can be insensitive’ (line 74)

Author Response

Response to in-house editor

We greatly appreciate your thoughtful comments.

  • Please include only relevant data in the tables. Any numbers that are not directly relevant to the study findings can be removed or put in the supplemental tables.
  • We removed variables in table 3,4 and 5; Long op. duration (≥300 min), Blood loss (≥400 mL), High ASA (≥3).

  • Figure 2- figures can be labeled as figure 2A and figure 2B.
  • We modified the figures with label.

  • Please make sure to avoid odd phrases such as ‘smooth treatment’ (line 73), ‘clinicians can be insensitive’ (line 74)
  • . We modified the phrases. “In most cases, clinicians pay careful attention to glucose control in patients with diabetes. However, in patients without DM, clinicians respond passively in SGL change.”.

Reviewer 2 Report

The authors successfully demonstrated that controlling post-operative glycemic status could prevent post-operative complications in patients with pancreatic surgery. As the authors pointed out, this study is a retrospective investigation which limits generalization. A future study with a prospective study defining post-operative glycemic control is encouraged.

1) Could you please show the detailed procedures for propensity score matching? Indicate the variables to match between the two groups.

2) Also, describe the selection procedures on multivariate logistic regression analysis.

3) The pre-operative ASA score shows a significant difference in Table 1 despite the description in Lines 191 to 192.

Author Response

  • Could you please show the detailed procedures for propensity score matching? Indicate the variables to match between the two groups.
  • We modified the result “We performed 1:1 propensity score matching to balance the differences in variables (Sex, age, hypertension and operation type) between two groups”

  • Also, describe the selection procedures on multivariate logistic regression analysis.
  • We modified the result “We select variables that universal risk factors of major complication and main outcomes of this study.”

  • The pre-operative ASA score shows a significant difference in Table 1 despite the description in Lines 191 to 192.
  • We modified the description. “except for age and pre-operative ASA score.”
